# Cell Therapy as a Way to Increase the Effectiveness of Hematopoietic Stem Cell Transplantation

**DOI:** 10.3390/cells13242056

**Published:** 2024-12-12

**Authors:** Ekaterina Pashkina, Elena Blinova, Maria Bykova, Alina Aktanova, Vera Denisova

**Affiliations:** 1Research Institute of Fundamental and Clinical Immunology, 14, Yadrintsevskaya st., 630099 Novosibirsk, Russia; blinovaelena-85@yandex.ru (E.B.); maria18021997@mail.ru (M.B.); aktanova_al@mail.ru (A.A.); verden@bk.ru (V.D.); 2Department of Clinical Immunology, Novosibirsk State Medical University, 52, Krasny Prospect, 630091 Novosibirsk, Russia

**Keywords:** cell therapy, immune recovery, hematopoietic stem cell transplantation

## Abstract

Hematopoietic stem cell transplantation (HSCT) is a standard method for treating a number of pathologies, primarily blood diseases. Timely restoration of the immune system after HSCT is a critical factor associated with the development of complications such as relapses or secondary tumors and various infections, as well as the graft-versus-host reaction in allogeneic transplantation, which ultimately affects the survival of patients. Introduction into the recipient’s body of immune system cells that are incapable of sensitization by recipient antigens during the period of immune reconstitution can increase the rate of restoration of the immune system, as well as reduce the risk of complications. This review presents the results of studies on cell therapy with various cell subpopulations of both bone marrow and mesenchymal origin during HSCT.

## 1. Introduction

Hematopoietic stem cell transplantation (HSCT) has evolved over the course of its history from a method that was declared ineffective and of limited applicability in the 1960s to a standard treatment for fatal malignant and benign blood diseases [1]. To date, approximately 1.5 million HSCTs have been performed in more than 1500 transplant centers worldwide. Timely restoration of the immune system after hematopoietic cell transplantation is a critical factor associated with the development of complications such as relapses or secondary tumor diseases and various infections, as well as graft-versus-host disease (GVHD) in allogeneic transplantation, which ultimately affects patient survival [2,3]. Protection against infectious agents and antitumor control primarily depend on the reconstitution of the T-cell component of the immune system.

Recovery of various immune cell subsets after HSCT occurs at distinct time intervals. [4]. After conditioning therapy, patients undergo an “aplastic phase” (severe neutropenia or pre-engraftment phase) until neutrophils are restored. The doses of total nucleated cells (TNCs) and CD34^+^ cells in the graft source are key factors influencing the engraftment rate and outcome following HSCT [5,6].

Myelopoiesis is restored first, followed by natural killer (NK) cell numbers within 21–100 days, while adaptive immunity takes longer to recover [7,8]. The process of T-cell pool restoration is divided into two independent pathways: thymus-dependent and thymus-independent [9,10], with the second pathway being predominantly restored by homeostatic proliferation of T-lymphocytes, which results in a contraction of T cell receptor repertoire diversity. In the early post-transplant period, lymphocyte restoration occurs by the thymus-independent pathway, that is, by mature donor T-cells that were transfused to the recipient along with hematopoietic stem cells during transplantation, and to a lesser extent by the expansion of previously existing naive T-cells and memory T-cells of the recipient that “survived” after the conditioning [9]. The thymus-dependent pathway involves the de novo formation of naive T cells in the thymus and the subsequent formation of a pool of memory T cells that implement the main immunological reactions—“graft versus tumor” and “graft versus host” [11,12]. Restoration of the T-cell link is key to protection against viral infections, which are a serious complication after GVHD.

The B cell compartment, which is involved in humoral immunity, is the slowest to recover [13] and can take as long as 5 years after allogeneic HSCT. Transient CD19^+^CD21^low^CD38^high^ B cells are the first B cells to migrate from the bone marrow, and their levels increase in the peripheral blood in the first months after HSCT, after which their percentage gradually declines, while the proportion of more mature B cell subsets increases [14]. The absence of CD19^+^CD27^+^ memory B cells, reduced levels of circulating immunoglobulins, impaired immunoglobulin class switching, and loss of complexity in immunoglobulin gene rearrangement patterns make patients undergoing allogeneic HSCT susceptible to encapsulated bacteria like *Streptococcus pneumoniae* and *Haemophilus influenzae*.

A promising approach to accelerate immune reconstitution and reduce risks in HSCT is cell therapy. Currently, various types of cell therapy schemes used in HSCT are being considered, including the introduction of cells before this therapy, co-transplantation when introducing stem cells into the recipient’s body, as well as the adoptive transfer of various cells, both once and in the form of a course with different durations and the introduction, of cells at different stages of immune system restoration.

## 2. Platelet and Red Blood Cell Therapy and HSCT

After chemotherapy and radiation therapy to prepare the patient for hematopoietic stem cell transplantation, pancytopenia is observed. It is known that rapidly dividing cells are most sensitive to the effects of cytostatic therapy, but high doses of drugs in the conditioning mode significantly reduce the number of formed elements of the blood. Patients who have received a hematopoietic stem cell transplant require extensive transfusion support with formed blood elements until the graft is established [15].

Platelets and erythrocytes, in addition to their direct functions of hemostasis and oxygen delivery to tissues, respectively, have immunomodulatory functions. It has been established that platelets participate in anti-infective protection, recruitment and stimulation of innate cell functions, activation of dendritic cells, and enhancement of the antigen presentation and adaptive immune response [16]. Depending on the microenvironmental conditions, erythrocytes may either facilitate immune activation or preserve immune quiescence [17]. Thus, platelets and erythrocytes may play an important role in modulating post-transplant immunity.

Platelet and red blood cell transfusion support are positively correlated with plasma tumor necrosis factor-alpha (TNF-α), interleukin-6 (IL-6), and soluble thrombomodulin levels [18] and do not increase the risk of developing aGVHD. The only predictable complication in the form of rejection reactions is the transfusion of red blood cells with platelets, incompatible according to the ABO system, to patients [19]. However, in the case of transplantation of certain organs rather than HSCT, the transfusion of red blood cells at an early stage is associated with the risk of developing rejection reactions [18]. Platelet accumulation in grafts contributes to the development of vasculopathy and is associated with the risks of acute and chronic graft rejection. This is facilitated by platelet-derived humoral factors and the expression of surface markers such as CD39 and CD154 and secreted chemokines such as PF4 and β-thromboglobulin (β-TG), which recruit neutrophils, and regulated on activation, normal T cell expressed and secreted (RANTES), which in addition to recruiting monocytes, also recruits T cells [20]. It should be noted that such a negative side of immunomodulation with these cells is associated with the transplantation of parenchymatous organs. And in patients with allotransplantation of stem cells in the early post-transplantation period, platelet transfusion did not cause transplant rejection reactions [21].

We summarized the various data on the side effects of red blood cell and platelet transfusion after HSCT (Table 1). Transfusion support for HSCT is an essential part of supportive care, and platelets and red blood cells should be transfused into recipients. Transfusion support is necessary to prevent the development of bleeding and anemia. The formation of red blood cells and megakaryocytes as a result of de novo hematopoiesis plays an important role in creating a microenvironment for stem cells and, consequently, better engraftment of the cells.

## 3. Granulocyte Transfusion

The granulocyte group includes neutrophils, basophils, eosinophils, and mast cells [26]. Neutrophils are the largest type of leukocytes in the blood of adult humans and are part of the innate immune system, helping to eliminate fungi and bacteria. Neutrophils, like other granulocytes, are formed quite quickly and have a short lifespan [27], due to which their pool is renewed faster than other white blood cell compartments. Because their numbers are easily restored, neutrophils have time to recover after HSCT within the week.

Neutrophils are also instrumental in regenerating the injured vascular niche and may provide strategies to accelerate regeneration of patients undergoing HSCT [28]. Newly generated neutrophils may promote regeneration of the sinusoidal network, which in turn facilitates the engraftment of hematopoietic progenitors. This positive vicious cycle continues until the vascular niche is restored and the bone marrow returns to homeostasis [29]. Depletion of mature neutrophils from the initial graft or genetic ablation of donor-derived neutrophils delayed regeneration of the vasculature in the bone marrow. Currently, there is evidence of the heterogeneity of neutrophil populations and their active participation in many processes associated with the activation of innate and adaptive immunity after HSCT [29].

It can be noted that granulocyte transfusion after HSCT may be effective in case of delayed recovery of neutrophil numbers. When evaluating the effectiveness of this type of cell therapy, ambiguous results were obtained [26]. This may be due to the insufficient number of patients in the study or the heterogeneity of the neutrophil pool. In a retrospective analysis, patients who received a mean transfusion dose ≥ 0.6 × 10^9^ granulocytes per kilogram tended to have better outcomes than those who received a lower dose [30]. Another study also showed that granulocyte transfusion was effective in only a small part of patients [31].

Since granulocytes are difficult to separate from erythrocytes, they contain a large number of donor red blood cells, and therefore cells for transfusion must be ABO-compatible with the recipient plasma to avoid hemolytic reactions [32]. In addition, the product must be irradiated to prevent acute graft-versus-host disease. Cellular products from donors must be transfused within 24 h. It is important to note that granulocyte infusions are associated with fever and may cause more serious reactions such as acute lung injury. Thus, the question of the necessity of using donor granulocyte transfusions and methods of increasing the safety and effectiveness of the procedure remains open.

## 4. Adoptive Transfer of Lymphocytes

### 4.1. Depletion of αβT-Lymphocytes and CD19

Due to the decreased ability of the body to resist infectious processes after HSCT, a number of studies have been conducted on the effectiveness of the adoptive transfer of donor lymphocytes [33,34,35,36]. As a result of αβ T and CD19 depletion, a cellular product with a high content of stem cells, committed myeloid precursors, NK cells, and γδ T lymphocytes is formed. This type of depletion prevents the possible presence of adaptive immune cells, which is safe and associated with a minimal risk of GVHD, as well as other immune and toxic complications. The use of low-dose infusions of CD45RA-depleted donor lymphocytes correlates with a decrease in the cumulative probability of transplant mortality and an increase in the overall survival of allogeneic HSCT recipients with αβ T and CD19 depletion of the graft, which is apparently associated with a reduced risk of severe viral infections. However, reliable data confirming a reduction in the risk of recurrence of oncohematological diseases and transplant rejection events have not been obtained. Of interest is also the use of individual populations of lymphocytes that are part of TCRαβ/CD19 cell–depleted cells, such as NK cells, helper ILCs, gamma-delta T cells, etc.

### 4.2. NK Cells

NK cells are one of the populations of immune system cells that perform immunological surveillance of the tumor and are responsible for killing tumor cells [37]. At the same time, NK cells recognize the absence of “self” MHC molecules and, as a result, have alloreactivity, participating in the GVHD reaction in alloHSCT. Tolerance of NK cells to their own tissues is formed in the process of the interaction of inhibitory killer-cell immunoglobulin-like receptor (KIR) with their own HLA I molecules, which excludes the development of autoreactivity [38]. In contrast, when the expression of HLA I molecules is impaired, for example, on cells damaged due to infection or tumor cells, signals activating KIR prevail, and the mechanism of destruction of these cells is launched. This property of NK cells formed the basis of the “missing-self” hypothesis [37].

After allo-HSCT, most NK cells have on their cell membrane a donor repertoire of KIR molecules, which differs significantly from the repertoire of recipient NK cells before allo-HSCT [38]. Therefore, NK cells expressing donor KIRs may exhibit alloreactive properties in recipients. However, an increase in the pool of alloreactive NK cells in the peripheral blood of recipients allows the development of graft-versus-tumor disease early after allo-HSCT [39]. One possible explanation for the occurrence of graft-versus-tumor disease is that transfusion of donor HSCT may create a transient dominant environment of donor HLA in the recipient bone marrow, and donor NK cells expressing KIRs may be “educated” by donor HLA, which affects their functions [40]. After leaving the bone marrow and migrating into a recipient-dominated environment, NK cells eventually lose their ability to respond to recipient HLA I molecules.

A study in mice showed that adoptive transfer of IL-2-activated natural killers together with donor hematopoietic stem cells promotes effective graft engraftment and reduces the risk of severe GVHD [38,41,42]. Moreover, transfusion of mice with Ly49^+^NK cells from MHC I-incompatible animals (Ly49 receptors in mice are similar to KIR in humans and recognize MHC class I molecules) promotes the elimination of residual host tumor cells and protects against GVHD by depleting host APCs [43]. In contrast, mice that received bone marrow transplantation without NK cell transfusion died from GVHD, and transfusion of NK cells from MHC I-compatible donors did not provide protection against GVHD. Subsequent studies also showed that donor alloreactive NK cells suppressed the development of GVHD by inhibiting T cell proliferation and activation [42,44]. A study in patients showed that patients with refractory AML who underwent haploidentical donor allo-HSCT (haplo-HSCT) followed by donor NK cell transfusion had a lower risk of developing GVHD than patients without NK cell transfusion [45]. The “protective” role of NK cells in the pathogenesis of GVHD has been questioned; in another study [46], patients who received transfusion of donor IL-15/4-1BBL-activated NK cells after haplo-HSCT with T cell depletion had a higher risk of GVHD. However, in phase I clinical trials, IL-2-activated NK cells from haploidentical donors demonstrated their safety and association with a low risk of developing GVHD [47].

Unlike alloreactive NK cells, autologous NK administration is not associated with the risk of GVHD, allowing for long-term infusion of these cells into patients without immunosuppressive conditioning [48]. Potential limitations of autologous NK cell-based products include the apparent lack of finished products and the ongoing manufacturing costs of producing the cell product. It has been suggested that autologous NK cell-based therapy may be particularly suitable for patients with low tumor burden, such as minimal residual disease. This is based on earlier studies in experimental model systems demonstrating the ability of NK cells to preferentially target small tumors [49].

An interesting option for cell therapy after auto-HSCT is the introduction of NK cells obtained from the umbilical cord blood of donors who do not match either the MHC I or KIR receptors [50]. A pilot study showed the safety of the therapy and a reduced risk of relapse in patients compared to the control group.

Thus, NK cell therapy after HSCT is primarily aimed at reducing the risk of tumor recurrence in both allo- and auto-HSCT (Table 2). In the case of GVHD prevention in allo-HSCT, the role of NK cells is controversial and requires further research.

### 4.3. Innate Lymphoid Cells (ILC)

ILCs were originally considered T-cell analogues, and non-cytotoxic ILCs were largely mirroring functions of T helper (Th) subsets, subdivided into ILC1, ILC2, and ILC3, and duplicating the functions of Th1, Th2, and Th17 cells, respectively [51]. However, it is now believed that ILCs are not only innate T-helper cell analogues but also participate as a link between sensory cells that monitor any changes in the cell environment that are not necessarily pathogenic and instruct effector cells that act to maintain tissue homeostasis [52]. Therefore, restoration of the ILC pool after HSCT is essential for immune system function. It was found that, unlike NK cells, helper-like ILCs recovered slowly, mirroring the pattern observed for T cells, with normalization achieved at 1 year [53,54]. Interestingly, the total absence of ILCs was not linked to an increased susceptibility to diseases, suggesting that in the context of modern medical care and hygiene, along with the presence of a functional adaptive immune system, ILCs may be redundant [55].

The mechanisms behind ILC reconstitution following depletion remain unclear. Hematopoietic stem and progenitor cell (HSPC) grafts comprise both progenitor and mature ILC [56]. It is likely that circulating ILCs originate from common lymphoid progenitor (CLP) cells in the bone marrow, as well as from mature ILCs that were co-transfused with the graft. ILCs are generally considered to be tissue-resident cells reliant on local self-renewal. However, since donor-derived ILCs have been observed in non-lymphoid tissues following allogeneic HSCT, it is plausible that the seeding of ILCs from the circulation also contributes to this process. Lastly, environmental factors can influence ILC plasticity.

Allo-HSCT, sourced from the blood of G-CSF-mobilized healthy donors, exhibited a frequency of ILC closely resembling that found in the peripheral blood of healthy individuals [57]. The composition of ILC subsets was also comparable to that in the blood of donors, with the notable exception of NKp44^+^ ILC3, which were significantly more prevalent in the HSCT grafts. Furthermore, the relative ILC content in the graft showed a tendency to correlate with ILC reconstitution following allogeneic HSCT, indicating that the peripheral expansion of transplanted mature ILC may play a role in the early reconstitution of ILC post-transplantation. Patients who received a graft with a relatively low ILC content faced a significantly higher risk of developing acute GVHD compared to those who received allogeneic HCT grafts that were relatively high in ILC [58]. In another study it was found that graft-versus-host disease (GVHD) correlated with reduced levels of activated and gut-homing NKp44^+^ innate lymphoid cell progenitors (ILCPs), which supports the notion that this specific ILC subset plays a crucial and non-redundant role in preventing this life-threatening condition under lymphopenic circumstances [54].

Donor ILC2 infusion successfully reduced the lethality associated with aGVHD and effectively treated lower gastrointestinal tract disease in murine models of allogeneic HSCT [58,59]. However, another study demonstrated that ILC2 may change their phenotype following transplantation due to plasticity, which is one of several possible mechanisms behind the poor reconstitution of these important cells following allo-HSCT [60].

In the case of ILC, there is a shortage of cells after transplantation, but this does not lead to an increase in the incidence of infectious complications or the risk of relapse but increases the chances of developing GVHD. ILC infusions have reduced the risk of developing GVHD in mouse models of allo-HSCT.

### 4.4. γδ T Cells

γδ T-lymphocytes are T cells, but unlike conventional αβ T-cells, they occupy an intermediate position between innate and adaptive immunity [61]. γδ T cells are known to have antitumor activity and are capable of antigen presentation and clonal expansion but have no MHC restrictions. According to the literature, γδ T cell reconstitution occurs within the first few weeks after HSCT [62,63,64,65]. In children with haploidentical HSCT who received αβ T cell-depleted grafts, γδ T cells rapidly expand to reach up to 90% of the initial T cell pool and subsequently decline as αβ T cells begin to reconstitute [66]. Further characterization of γδ T cell composition one month after therapy revealed that the γδ T cell composition was not significantly different from that present in the peripheral blood of donors or healthy adults [66]. Similarly, studies of γδ T cell reconstitution at the clonal level showed that the reconstituted γδ T cell repertoire in blood remained highly stable over time after HSCT [65].

Clinical trials assessing the use of γδ T cells for targeting hematologic malignancies can be categorized into three distinct groups: (a) in vivo stimulation of autologous γδ T cells, (b) adoptive transfer of ex vivo expanded autologous γδ T cells, and (c) adoptive transfer of ex vivo expanded allogeneic γδ T cells [67]. As a result, an increase in survival and a decrease in the risk of GVHD were observed in most studies. However, Pabst et al. showed that patients who received higher levels of donor γδ T-lymphocytes experienced a higher incidence of acute GVHD (66% versus 40%, *p* = 0.02) [68]. Presumably, differences in cell effects may be due to different γδ T cell subsets, which must be taken into account when conducting therapy.

### 4.5. T Regulatory Cells (Treg)

It is known that Tregs provide tolerance to both auto- and alloantigens [69,70,71]. A significant decrease in the Treg:Th cell ratio at 2 weeks after allo-HSCT transplantation is associated with the development of acute GVHD [72], and a high probability of developing acute GVHD was also observed in patients with a low number of granzyme B^+^ Treg [73]. During autologous HSCT, an increased content of CD4^+^Foxp3^+^ cells (6.7%) in the peripheral blood of patients with multiple myeloma on the day of graft engraftment led to a relapse or progression of the tumor disease within 12 months after transplantation [74], which is possibly associated with the suppression of T-effector cell expansion at an early stage of immune reconstitution, while no cases of acute and chronic GVHD were recorded. The development of GVHD is also affected by the number of Tregs in the graft: a significant decrease in the likelihood of developing GVHD after allo-HSCT is observed with a Treg content of 10 × 10^6^/kg or more [75,76].

The suppressive effect of Tregs on GVHD is provided by contact-dependent and contact-independent mechanisms: cytolysis of effector cells (secretion of perforin and granzyme B); secretion of IL-10, TGF-β, and IL-35 immunosuppressive cytokines; depletion of IL-2 due to its greater consumption by Tregs due to high expression of CD25; induction of enzymatic breakdown of adenosine triphosphoric acid to adenosine; modulation of DC function both at the maturation stage and during the implementation of the antigen-presenting function [77].

It has been shown in animal models that the infusion of freshly isolated or ex vivo-obtained Tregs in various modes (prophylactically—before HSCT; therapeutically—simultaneously with the transplant or the day after transplantation) can restrain the development of GVHD or even prevent its development and ensure early immunoreconstitution [78,79]. The duration of the effect in preventing GVHD is directly related to the expansion and survival of Tregs in the recipient’s body [78]. The adoptive transfer of even low doses of Tregs can prevent the development of GVHD while maintaining the cytotoxic activity of conventional T cells in the graft-versus-tumor reaction [78]. However, there is evidence that the infusion of induced Tregs can reduce the effectiveness of the antitumor response after transplantation and lead to a short-term relapse of leukemia [80,81].

Most clinical trials involving adoptive transfer of Tregs in patients with hematological malignancies are aimed at reducing the incidence or severity of GVHD after allogeneic HSCT. The effect in preventing GVHD was confirmed using freshly isolated donor Tregs [82,83], which were administered to patients 4 days before HSCT. Repeated administration of in vitro expanded cord blood Tregs (0.1–30 × 10^5^ cells), starting from the day after transplantation and until the absolute neutrophil count was above 2500/μL, reduced the incidence of acute GVHD (43% versus 61% in the historical control without Tregs) and did not increase the risks of infections, disease relapse, or early mortality [84]. Administration of ex vivo expanded CD4^+^CD25^+^CD127^−^ Tregs to patients with chronic GVHD significantly alleviated symptoms and reduced immunosuppressive pharmacotherapy [85]. However, no convincing effect of adoptive Treg therapy in the treatment of acute GVHD was demonstrated [86]. In 4 out of 5 patients with chronic GVHD refractory to treatment and receiving Tregs from HLA-matched donors after transplantation, an increase in Treg in the peripheral blood was observed against the background of a decrease in the amount of immunosuppressive therapy received, and 2 out of 5 patients achieved relief of GVHD symptoms [87]. However, the use of Tregs for the treatment of GVHD cannot be called successful, since skin cancer was diagnosed in 2 patients several months after the therapy [87].

Adoptive therapy uses not only Tregs but also added conventional donor T cells, which allows for greater success in preventing GVHD. Thus, within the HLA-haploidentical HSCT protocol, including the introduction of donor Tregs and infusion of donor T lymphocytes, the probability of relapse-free/moderate/severe chronic GVHD was 75% with a median follow-up of 29 months [88]. Patients with acute myeloid leukemia (50 people) received an infusion of 2 × 10^6^/kg donor Tregs on day −4, then 1 × 10^6^/kg donor T cells on day −1, and an average of 10.7 × 10^6^ ± 3.4 × 10^6^/kg CD34^+^ HSCs on day 0. Moreover, after transplantation, leukemia relapse was observed in 4% of patients (2 people); 33% of patients (15 people) developed acute GVHD without transition to chronic, 2 of 15 patients died from acute GVHD, and 1 of 15 patients died from leukemia relapse [88].

In addition to classical CD4^+^CD25^+^Foxp3^+^ Treg, Tr1 cells (CD4^+^Foxp3^−^LAG3^+^CD49b^+^) that produce high levels of IL-10 are capable of supporting the restoration of immune cells and preventing the development of GVHD [89]. In 4 out of 12 patients with hemoblastoses, after the introduction of alloantigen-specific donor IL-10-anergized T cells (IL10-DLI) containing Tr1 (on average 35 days after haplo-HSCT), GVHD did not develop, and disease remission was maintained for a long time [90].

Thus, Tregs play an important role in limiting immune responses associated with alloantigen recognition and in preventing GVHD while maintaining an antitumor response against residual tumor clones and without increasing the risks of disease relapse and post-transplant infections. However, the use of Tregs is associated with a number of difficulties. The number of Treg populations in the body is small and, according to various estimates, is 4–10% of CD4^+^ T lymphocytes in peripheral blood; to ensure the required number for transplantation, it is necessary to resort to ex vivo Treg expansion [91], which in turn can affect the functional state of the cells. When introduced into the body, the question arises about the stability of Tregs, especially induced Tregs, which are highly plastic. Tregs are capable of acquiring various transcriptional programs in response to changing environmental conditions, in particular through cytokine signaling [92], and losing their immunosuppressive function. A solution to this problem may be Treg cells modified using genetic engineering approaches that maintain stable expression of Foxp3, IL-10, the ability to induce tolerance to alloantigens, and even antitumor activity [93,94,95,96,97]. Results of Treg cell therapy are presented in Table 3.

## 5. Co-Transplantation of Mesenchymal Stem Cells

Mesenchymal stem cells (MSCs) are fibroblast-like cells capable of differentiating into various cell types, such as osteoblasts, chondrocytes, adipocytes, and others [98]. Characteristic features of MSCs are their adhesive ability to plastic and the expression of CD73, CD90, and CD105. MSCs can be obtained and efficiently collected from various tissue sources without significant ethical concerns, which determines their convenience for use in cell therapy [99].

A number of functional properties of MSCs suggest beneficial effects from the co-transplantation of these cells in HSCT. First, as key players in the bone marrow niche, MSCs support hematopoietic stem cell (HSC) function by secreting multiple factors (SCF, angiopontin, VCAM1, osteopontin, and extracellular matrix components) and expressing CXCL12 [100,101]. Second, MSCs do not stimulate alloreactivity and are not lysed by cytotoxic T cells and natural killers. Third, MSCs exhibit strong immunosuppressive properties, such as inhibition of activated T cell proliferation; downregulation of IL-2, TNF-α, IL-1β, and IFN-γ secretion; and stimulation of regulatory transcriptional complex expression, which supports Treg formation [99]. This suppressive activity has been shown to underlie the reduction in the risk of alloreactivity development during MSC co-transplantation in the context of HSCT [102]. Thus, the ability to modulate the bone marrow microenvironment, safety, low immunogenicity, and immunosuppressive properties of MSCs make them an interesting candidate for improving HSCT outcomes. Indeed, clinical studies demonstrate the effects of MSC co-transplantation in the form of a reduced risk of GvHD development and rapid restoration of hematopoiesis in the treatment of pathologies of various origins, including blood cancer [101,103,104].

Interestingly, MSCs, in the case of hematological malignancies, are involved in enhancing the engraftment efficiency of both myeloid and lymphoid lineages. In a study investigating the effect of co-transplantation of autologous MSCs on the early and late (homeostatic proliferation and thymopoiesis, respectively) stages of restoration of different T-cell subsets in patients with hematological malignancies, an improvement in the early restoration of both naive and memory cells was demonstrated, with a more pronounced effect on CD4^+^ lymphocytes in the MSCs(+) group compared to MSCs(-). The authors of the study emphasize that despite their immunosuppressive properties, under certain conditions, namely a small number of MSCs or a low basal level of T-lymphocyte proliferation, MSCs are capable of stimulating T-lymphocytes. At the same time, according to the results of the experiments, the restoration of CD4^+^ in the early post-transplant period was associated with the anti-apoptotic effect of MSCs. In addition, it was shown that patients from the MSCs(+) group were distinguished by more effective restoration of early thymic migrants (RTEs) [105,106]. Other studies by the same group of authors also showed earlier and more effective restoration of lymphocytes for patients with co-transplantation of auto-MSCs [107]. Thus, it is obvious that co-transplantation of auto-MSCs has an advantage in more pronounced restoration of the T-lymphocyte link compared to standard HSCT, both at the early and late post-transplantation stages.

Promising results were also obtained in a study of myeloid lineage cell regeneration during co-transplantation of auto-MSCs together with HSCs for blood cancer therapy. A significant reduction in the period of neutropenia and thrombocytopenia was shown. Interestingly, this result did not depend on the number of administered MSCs [108]. The above data are consistent with the results of other phase I/II clinical studies, which demonstrated a higher frequency of neutrophil and platelet engraftment in groups with pre-incubation of HSCs with allo-MSCs before administration. The authors associate these results with the ability of MSCs to repeat some signals of the bone marrow niche, which increases the proliferation and survival of progenitor cells in vitro before administration [109,110]. Thus, the use of MSC co-transplantation in both allo- and autotransplantation demonstrates positive effects for the engraftment of both myeloid and lymphoid lineage cells, indicating the prospects for their use in the context of HSC transplantation for hematological malignancies.

However, there is evidence of negative effects of MSCs in this context. Thus, in a clinical study assessing the effect of MSCs co-transplantation in allo-HSCT for people with hematological malignancies, an increased incidence and rate of relapse was shown for the MSCs(+) group compared to the MSCs(-) group. This was explained by the suppression of the graft-versus-tumor (GvT) response due to the immunosuppressive properties of MSCs. Moreover, the results of this study did not confirm the role of MSCs in improving the restoration of hematopoiesis parameters. In this regard, the authors suggest that MSC co-transplantation in allo-HSCT is more applicable in the treatment of non-malignant diseases, but not malignant hematopoietic diseases [111]. This example demonstrates that despite the numerous advantages of MSCs, there are certain risks associated with their use. Perhaps the discrepancies in the results may be associated with the protocol of the procedure and the characteristics of the presented sample. Indeed, some preclinical studies demonstrate that the results of cell therapy are affected by such parameters as the mode of radiation or chemotherapy, the dose of administered MSCs, and the conditions of administration of both MSCs and HSCs [112].

To summarize the above, it is necessary to pay attention to the remarkable prospects for the use of MSC co-transplantation in HSCT for leukemia treatment. Nevertheless, such studies require a certain degree of caution and the development of optimal therapy protocols.

## 6. Effect of Cell Therapy on Risks That Reduce the Effectiveness of HSCT

### 6.1. Transplant Failure

HSCT is associated with many complications, including those associated with post-transplant pancytopenia. Recent studies defined poor graft function (PGF) as at least bilinear severe cytopenia, which is characterized by the following: (1) ANC ≤ 0.5 × 10^9^/L, (2) platelet count ≤ 20 × 10^9^/L, and (3) hemoglobin ≤ 70 g/L for at least 3 consecutive days beyond day +28 after HSCT or the requirement of transfusion with full chimerism and a hypoplastic-aplastic bone marrow, while severe graft versus host disease (GVHD) and relapse were excluded [113]. Therefore, the rate of recovery is directly related to the development of complications and survival after HSCT. The co-transplantation of mesenchymal stem cells can increase the rate of hematopoiesis recovery [114]. However, the introduction of a number of other cells, such as erythrocytes [18], platelets [21], and neutrophils [28], helps to fill empty niches and ensures the maintenance of the immune system in the periphery during the period of pancytopenia.

### 6.2. Relapse of Cancer

Relapse is the most common cause of treatment failure after HSCT [115]. Antitumor activity is predominantly exerted by cytotoxic T-lymphocytes and NK cells; infusions of killer cells can be used in the relapse of leukemia [34,35,36,41,42,43,44,45,46,47,48,49]. However, the use of these cells is associated with a number of complications, including the development of GVHD. It is also possible to use γδT cells to enhance the antitumor response [67]. Cell therapy is still facing challenges regarding the difficulties in its expansion and the standardization of protocols.

### 6.3. Graft-Versus-Host Disease (GVHD)

GVHD is one of the most common complications in allo-HSCT [116]. The main drugs for its prevention remain calcineurin inhibitors (cyclosporine A and tacrolimus), methotrexate, and mycophenolate mofetil. With the spread of low-intensity conditioning regimens, antithymocyte globulin has become widely used. However, a “negative” effect on the reconstitution of the T-cell link of immunity was noted, which increases the risk of severe infectious complications and relapse of the disease. With the increasing number of alternative (partially matched and haploidentical) donor transplants, cyclophosphamide (CP) has been widely used as a prophylactic agent. Adoptive transfer of NK cells, γδT cells, ILCs, and T-regulatory cells may be considered additional cell therapy.

### 6.4. Infections

First of all are viral infections such as cytomegalovirus (CMV) and Epstein–Barr virus (EBV), with viral reactivation thought to reflect the status of T cell recovery [117]. Memory CD8^+^ T cell populations seem to play a crucial role in providing protection against CMV and EBV or in their elimination, while for other herpes viruses like adenovirus and HHV-6, the count of CD4^+^ T cells is the primary predictor of both reactivation and outcomes. Additionally, CD4^+^ T cell count is a key predictor of long-term immunity against CMV. The reactivation of CMV significantly stimulates overall T cell recovery, with the most pronounced effect observed six months after HSCT. In contrast, reactivation of HHV-6 may lead to reduced T cell counts in patients at both six months and one year following HSCT. The adoptive transfer of antigen-specific T cells is anticipated to become more common in the near future, as this approach directly addresses the mechanisms of viral reactivation.

## 7. Conclusions and Future Perspectives

Cell therapy, both in the form of co-transplantation with HSCT and in the form of cell infusions before and after HSCT, is an important addition to this type of therapy. The use of various cell therapy options can prevent various types of complications observed in both auto-HSCT and alloHSCT. An important advantage of cell therapy in HSCT is the ability to combine various methods of both introducing different types of cells and conventional therapy, which is standardly used to reduce the risk of complications. We assume that in order to increase the effectiveness of HSCT, it is necessary to develop complex therapy based on infusions of different types of cells, carried out individually depending on the parameters of the recipient. The basis for a personalized approach can be information about the diagnosis, the presence of minimal residual disease, indicators of the immune system and cellular composition before therapy, the pattern of leukocyte recovery, parameters of immune reconstitution in dynamics after HSCT, etc.

One of the important areas of further research is the combination of HSCT with concomitant cell therapy and various modern methods of antitumor therapy, including monoclonal antibodies and cell therapy. An equally promising direction is the creation of cell products in which the genes of the main histocompatibility complex molecules I and II would be switched off, but at the same time, there would be signals for inhibiting the recognition of “lack of self”, which would allow avoiding a number of complications. It should also be noted that it is necessary to improve technologies that allow producing the necessary cells ex vivo, which will make this type of therapy more accessible.

## Figures and Tables

**Table 1 cells-13-02056-t001:** Complications after red blood cell and platelet transfusions after HSCT.

Type of Therapy	Treatment Regimen	Complications	References
Red blood cells	Between day −7 pretransplantation and day +27 post-transplantation, but excluding transfusions administered after a diagnosis of aGVHD; transfusion of 5 units of red blood cells on average (range 0 to 30).	A total of 239 patients (74%) of 322 developed aGVHD at day +150 post-transplant: grade I, *n* = 106 (33%); grade II, *n* = 70 (22%); grade III, *n* = 47 (15%); and grade IV, *n* = 16 (5%). Increasing the number of red blood cell transfusions was independently associated with a higher incidence of aGVHD and poor survival.	[22]
Platelets	From 0 to +60 days after allo-HSCT; transfusion of 5 units of platelets	The frequency of transfusion events was linked to a higher 100-day non-relapse mortality, extended post-transplant hospital stays, increased likelihood of requiring intensive care unit admission, and a greater number of organs impacted by severe toxicity.	[23]
Erythrocytes + platelets	From 0 to +13 after allo-HSCT; the median number of platelet and red blood cell transfusions received was 7 and 3 units, respectively	On day +100, 217 out of 664 patients (33%) developed grade II-IV aGVHD (average time to develop aGVHD was 24 days).	[24]
Erythrocytes + platelets	46.3% of patients after HSCT required red blood cell and platelet transfusions after day 30	Failure to achieve red blood cell (RBC) transfusion independence by day 30 was associated with poorer 5-year overall survival, decreased leukemia-free survival, and higher non-relapse mortality	[25]

**Table 2 cells-13-02056-t002:** Summary data on the efficacy and risk of complications of NK cell therapy in HSCT.

Type of Cells	Type of HSCT	Results of Cell Therapy	References
CD56^+^NK	Allo-HSCT	Absence of a GVHD or infections and improved restoration of the immune repertoire of T cells, Tregs, and NK cells.	[45]
IL-15/4-1BBL-activated CD56^+^NK cells	Allo-HSCT	increased risk of a GVHD (was observed in 5 of 9 patients)	[46]
IL-2-activated NK cells	Allo-HSCT	Decreased risk of a GVHD; out of the 16 treated patients, 11 were alive and in complete remission from hematologic malignancies without GVHD	[47]
Autologous NK cells	Auto-HSCTa	Patients showed evidence of objective measurable responses to NK cell infusions in the form of decreased M component and minimal residual disease.	[48]
NK from umbilical cord blood of donor	Auto-HSCT	It was a tendency of difference in decreasing the relapse rate between the group after NK cell infusions and the control group (9.7% vs. 24.4%).	[50]

**Table 3 cells-13-02056-t003:** Summary of data on Treg therapy for the prevention of complications of HSCT.

Type of Tregs	Therapy Regimen	Results of Cell Therapy	References
Donor Treg (freshly allocated)—1–2 × 10^6^/kg, 4 × 10^6^/kg; Tcons, donor (frozen before the G-CSF course)	Once. 4 days before HLA-haploidentical HSCT, Tcons (0.5–1.0 × 10^6^/kg) was transplanted with CD34^+^ cells (9.4 × 10^6^/kg).	In 2 of 28 patients, the transplant did not take root. 2 of 26 patients (7.7%) developed acute GVHD (≥grade 2) in those of 5 who received 4 × 10^6^/kg Treg and 2 × 10^6^/kg Tcons. Chronic GVHD did not develop in any patient during the follow-up period (mean 11.2 months (3.6–21.4 months)). There were no deaths caused by CMV (usually the mortality rate is on the order of 40% according to their previous data). During the first 2 months after transplantation, there were no fatal cases of infections. 13 of 26 patients (50%) died as a result of hepatic veno-occlusive disease (3 patients), multiple organ failure (1 patient), adenovirus infection (1 patient), adenovirus infection and GVHD (1 patient), GVHD (1 patient), bacterial sepsis (1 patient), systemic toxoplasmosis (1 patient), fungal pneumonia (3 patients), and CNS aspergillosis (4 patients). One patient with AML relapsed 6 months after transplantation due to that transplantation being performed with chemoresistant relapse from a non-NK-alloreactive donor. At a median 12-month and 21-month follow-up period, 12 of their 26 patients (46.1%) were alive and had no disease relapse.	[82,83]
Treg from cord blood donor #2 (in vitro expansion with beads coated with aCD3/aCD28 monoclonal ATs, 18 days)—1–30 × 10^5^/kg	Twice, after transplantation of donor umbilical cord blood cells #1 (≥3.0 × 10^7^/kg). The first time (1, 3, 10, or 30 × 10^5^/kg)—on the next day after HSCT, and the second time (30 × 10^5^/kg)—on the 15th day after HSCT	Acute GVHD (≥grade 2)—43% of patients on Treg versus 61% of patients without Treg, that is, Treg contributed to a reduction in acute GVHD. Two of the 14 (14%) patients with Treg at risk developed chronic GVHD; none of the patients who received a high dose of Treg (≥30 × 10^5^/kg) had chronic GVHD. What is significantly different from the historical control rate is that 26% of patients developed chronic GVHD. On the 100th day after HSCT, opportunistic infections were recorded in 9 of 23 patients (39%), and of these, 8 patients were on the immunosuppressive regimen of cyclosporine A + mycophenolate mofetil and 1 patient was on sirolimus + mycophenolate mofetil. Which is comparable to the historical control (108 patients) who received the same therapy without Treg co-transplantation; opportunistic infections developed in 53% of patients. In 8 of 23 patients (32%) treated with Treg, relapse of the disease occurred (2 with lymphoma, 4 with AML, and 2 with lymphoblastic leukemia); in historical controls, relapse occurred in 50% of patients. One-year recurrence-free survival was 41% of all patients and 58% of patients receiving high doses of Treg (≥30 × 10^5^/kg). Mortality from bacterial infection (3 patients), acute GVHD (1 patient), and multi-organ failure (1 patient) was comparable to historical controls.	[84]
Treg from donor cord blood #2 (in vitro expansion with modified K562 cells expressing CD64 and CD86—KT 64/86, aCD3 monoclonal AT, IL-2—18 days)—3–100 × 10^6^/kg	Once, after transplantation of cord blood cells of the donor #1 (≥3.0 × 10^7^/kg).	On day 100 after transplantation, acute GVHD (≥grade 2) developed in 9% of patients, which is significantly lower than in the control (45%, *n* = 22). One year later, no cases of chronic GVHD were recorded in any of the 11 patients treated with Treg; chronic GVHD developed in 3 patients (14%) in the control group. A total of 17 cases of infection were reported in 9 patients treated with Treg and 31 cases in 14 controls. On day 180 after HSCT, there was a significantly faster recovery of CD4^+^ cells and a population of naïve CD4^+^CD45RA^+^CCR7^+^ cells in patients treated with Treg compared to patients in the control group. Relapse occurred in 3 of 9 (33%) surviving patients treated with Treg and in 8 of 20 (40%) patients in the control group. After 6 months, 2 patients treated with Treg and 2 patients in the control group died; the cause of death was not related to disease recurrence. The one-year relapse-free survival rate was 55% in both the Treg-treated and control groups. The one-year overall survival rate in the Treg-treated group was 81%, and in the control group, 61%; however, no significant differences in this indicator were found between the groups.	[85]
Sorted donor Treg CD3^+^4^+^25^hi^CD127^−^ (in vitro expansion with inactivated autologous plasma and 1000 U/mL IL-2, beads coated with aCD3/aCD28 monoclonal ATs, 21 days)—1 × 10^5^/kg (chronic GVHD); 6 × 10^7^ Treg/per 1 administration or 3 × 10^6^/kg (acute GVHD)	Once, at 35 months after HSCT (HLA-compatible HSCT (sibling)) against the background of therapy (prednisone, tacrolimus, mycophenolate mofetil) with chronic GVHD (developed on the 137th day after HSCT). Three times, on days 75, 82, and 93, against the background of acute GVHD (developed on day 22 after HSCT) after allogeneic PBSCT from HLA-compatible sibling.	In a patient with chronic GVHD, after the administration of Treg, the patient’s condition improved and the amount of immunosuppressive therapy received decreased: complete withdrawal of mycophenolate mofetil and reduction in the dose of prednisone (5 mg/day; 0.1 mg/kg body weight). The number of CD4^+^Foxp3^+^ cells in the PC increased from 2.5% to 5% after 6 months. after the introduction of Treg. Serum levels of IL-6, IL-10, IL-7, and CCL21 decreased. In the case of acute GVHD, there was a slight (moderate) improvement in the patient’s condition after the first administration of Treg, which persisted after the subsequent administration of Treg. The patient died on the 112th day after HSCT from multiple organ failure. No changes in the level of Foxp3 cells and cytokines were recorded.	[86]
Sorted donor Treg CD4^+^CD25^high^ (in vitro expansion with 1000 U/mL IL-2 and 100 ng/mL rapamycin, beads coated with aCD3/aCD28 monoclonal AT (bead-to-cell ratio 1:1), 7–12 days)—0.97–4.45 × 10^6^/kg (mean 2.4 × 10^6^/kg)	Once, on average, 35 months after allogeneic HSCT (HLA-matched donor) during chronic GVHD therapy. In addition, 3 patients received low doses of IL-2—0.3 × 10^6^ IU/m^2^ daily (from 1 to 8 weeks and from 13 to 17 weeks after Treg administration)	2 out of 5 patients achieved relief from GVHD symptoms; the remaining 3 patients had stable symptoms of chronic GVHD up to 21 months. In 4 out of 5 patients who received Treg from HLA-matched donors after transplantation, an increase in Treg in the peripheral blood was observed against the background of a decrease in the amount of immunosuppressive therapy received. A few months after therapy, 2 patients were diagnosed with skin cancer (after 4 months, 1 patient was diagnosed with malignant melanoma; after 11 months, 1 patient was diagnosed with Bowen’s skin cancer (squamous cell carcinoma of the skin)). The median follow-up period was 19 months; 3 of 5 patients (60%) were alive. One patient died 8 months after administration of Treg from sepsis; one patient died after 4.5 months. After administration of Treg for acute hepatic failure. None of the patients had a recurrence of oncohematological disease, which indicates the absence of suppression of the graft-versus-tumor Treg reaction.	[87]
Treg, haploidentical donor (immunomagnetic separation of CD4^+CD25+^ cells before conditioning, CD4^+^CD25^+^CD127^−^Foxp3^+^ cell count 71 ± 8.5%) 2 × 10^6^/kg, T-cells, haploidentical donor (immunomagnetic separation of CD3^+^ cells prior to conditioning) 1 × 10^6^/kg	Once, 4 days before HSCT. 1 day before HSCT, donor T-cells (1 × 10^6^/kg) HLA-haploidentical HSCT (10.7 × 10^6^ ± 3.4 × 10^6^/kg CD34^+^ HSCT) were injected. Patients did not receive immunosuppressive therapy after HSCT.	The study included 50 patients with AML. After transplantation, leukemia recurrence occurred in 4% of patients (2 people) with a median follow-up of 34 months; 30% of patients (15 people) developed acute GVHD (≥grade 2) without becoming chronic. 2 of 15 patients died from acute GVHD, and 1 of 15 patients died from leukemia recurrence. Acute GVHD developed in 3 of 19 patients on TBI (whole body irradiation) conditioning and in 12 of 31 patients on TMLI (irradiation of the bone marrow and lymphoid organs—lymph nodes and spleen) conditioning. 4 patients developed moderate chronic GVHD, which completely resolved under the influence of therapy. One patient with severe chronic GVHD was recorded who subsequently died. The median follow-up was 29 months, during which the recurrence-free survival with moderate/severe chronic GVHD was 75%. The overall survival rate was 77%. Survival did not depend on the radiation protocol of patients. Sepsis was reported in 19 patients with 65 cases of febrile neutropenia. Two patients died as a result of septic shock caused by multidrug-resistant bacteria. In 22 of the 50 patients, CMV infection was detected, and antiviral drugs helped resolve the infection. No deaths associated with CMV infection were recorded. Recurrence-free mortality was recorded in 10 patients (20%). The causes of death were veno-occlusive disease (2 patients), aGVHD (2 patients), cGVHD (1 patient), septic shock (2 patients), unspecified pneumonia (1 patient), invasive aspergillosis (1 patient), and cerebral hemorrhage (1 patient).	[88]
Alloantigen-specific IL-10-anergized T cells are (IL10-DLI), donor cells obtained in a mixed culture of donor and host lymphocytes (CD3-depleted) in the presence of IL-10 for 10 days—cells must have low expression of CD25 and HLA-DR activation markers, be anergistic in the proliferation test with host alloantigens, and contain fewer CD8+ cytotoxic cells). 1 × 10^5^, 3 × 10^5^ (1 patient) CD3^+^ T-cells/kg	Once, on average 35 days after allogeneic HLA-haplo HSCT (12 × 10^6^/kg CD34^+^ HSCT)	The study included 17 patients (there were 19, but 2 did not undergo transplantation). It was possible to generate anergic IL10-DLI cells in 14 out of 17 patients; another 2 patients failed to inject cells due to transplant rejection and failure to achieve the criteria for early immunoreconstitution. obtained IL10-DLI cells. Among these patients (who did not receive IL10-DLI cells), 2 died from infections, 2 had a relapse of the disease, and 1 patient underwent a second transplant with cells from the same haploidentical donor due to transplant rejection after the first transplant. In 4 out of 12 patients, IL10-DLI cells were effective: the patients were alive on day 100 after transplantation, and immunoreconstitution occurred on day 28 after administration of IL10-DLI cells. Among the remaining 8 patients, 3 died from infections (43, 69, and 97 days after transplantation), and 4 patients experienced disease recurrence or transplant rejection. Another patient who received a high dose of IL10-DLI cells, despite early immunoreconstitution, developed severe GVHD that did not respond to therapy and died 1 year after transplantation. Administration of IL10-DLI cells promoted early immunoreconstitution (on average at 30 days, whereas recovery usually takes about 1 year).	[90]

## Data Availability

No new data were created or analyzed in this study.

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
