# Peer review of "Cell Therapy as a Way to Increase the Effectiveness of Hematopoietic Stem Cell Transplantation"

_cells, 2024, doi:10.3390/cells13242056_

Round 1

Reviewer 1 Report

Comments and Suggestions for Authors

This review explores the use of cell therapy to enhance the outcomes of HSCT. The authors discuss different cell types that could be used to improve immune system recovery and reduce complications like GVHD and tumor relapse. The review provides an in-depth analysis of different cell therapy approaches and addresses practical challenges, such as the immune suppression associated with some therapies, safety concerns, and the conflicting findings regarding their effectiveness in reducing tumor recurrence. The authors conclude that individualized complex therapies involving multiple cell types might be necessary to optimize HSCT outcomes.

The paper is comprehensive in its scope, covering a wide variety of cell therapy options to enhance the outcomes of HSCT. Its strength lies in the detailed descriptions of the different immune cell populations and their roles in both the innate and adaptive immune responses. Additionally, it effectively ties together preclinical and clinical findings, providing a balanced view of the state of cell therapies in HSCT.

However, the paper could have benefited from the following aspects:

  1. More structured comparative data regarding the effectiveness of each type of cell therapy across various patient outcomes. 
  2. A clearer separation between the advantages and limitations of each therapy type could provide more information for readers. 
  3. While the paper acknowledges conflicting results regarding the efficacy of certain cell types, more emphasis on why these conflicts arise, such as variability in patient profiles, conditioning regimens, or cell dose would have enhanced the reader's understanding. 
  4. The need for personalized treatment is a strong concluding point, but specific strategies for implementation could have been more explicitly outlined.

Author Response

This review explores the use of cell therapy to enhance the outcomes of HSCT. The authors discuss different cell types that could be used to improve immune system recovery and reduce complications like GVHD and tumor relapse. The review provides an in-depth analysis of different cell therapy approaches and addresses practical challenges, such as the immune suppression associated with some therapies, safety concerns, and the conflicting findings regarding their effectiveness in reducing tumor recurrence. The authors conclude that individualized complex therapies involving multiple cell types might be necessary to optimize HSCT outcomes.

The paper is comprehensive in its scope, covering a wide variety of cell therapy options to enhance the outcomes of HSCT. Its strength lies in the detailed descriptions of the different immune cell populations and their roles in both the innate and adaptive immune responses. Additionally, it effectively ties together preclinical and clinical findings, providing a balanced view of the state of cell therapies in HSCT.

However, the paper could have benefited from the following aspects:

More structured comparative data regarding the effectiveness of each type of cell therapy across various patient outcomes.

1. We have added tables for better presentation of  data.

A clearer separation between the advantages and limitations of each therapy type could provide more information for readers.

2. We have added some information about advantages and limitations of therapy

While the paper acknowledges conflicting results regarding the efficacy of certain cell types, more emphasis on why these conflicts arise, such as variability in patient profiles, conditioning regimens, or cell dose would have enhanced the reader's understanding.

3. We have made the corrections

The need for personalized treatment is a strong concluding point, but specific strategies for implementation could have been more explicitly outlined.

4. We have made the corrections and added information about possible strategies and research directions.

Reviewer 2 Report

Comments and Suggestions for Authors

Thank you very much for the opportunity to review the manuscript titled "Cell Therapy as a Way to Increase the Effectiveness of Hematopoietic Stem Cell Transplantation". It was an honor to contribute to this work, which addresses a highly relevant topic and demonstrates significant potential for publication. 

I have identified minor points for adjustment, such as providing more detailed discussions in specific sections, incorporating tables or figures to summarize key information, and refining the formatting for consistency. With these suggested revisions, I believe the article will fully meet the journal's standards and make a meaningful impact on the scientific community.

Minor Revisions

The text requires a review of acronyms, which should be written in full when first mentioned in the text. For example:

  • Line 38: natural killer (NK)

  • Line 51: graft-versus-host disease (GVHD)

  • Line 83: tumor necrosis factor-alpha (TNF-α), interleukin-6 (IL-6)

  • Line 92: regulated on activation, normal T cell expressed and secreted (RANTES)

  • Line 159: killer-cell immunoglobulin-like receptor (KIR)

This adjustment should be applied consistently throughout the manuscript.

Major Revisions

I believe the authors could enhance the manuscript by:

  1. Adding Tables/Graphs: Summarize key data (e.g., success rates of different cell therapies) in tables or graphs to improve visualization, particularly for topics like the use of NK cells.

  2. Including a Future Perspectives Section: Add a specific section outlining challenges and future directions to enhance the clinical applicability of the findings. This would provide valuable insights for advancing the field.

With these revisions, the manuscript would further solidify its contribution to the literature and improve its clarity and impact.

Author Response

Thank you very much for the opportunity to review the manuscript titled "Cell Therapy as a Way to Increase the Effectiveness of Hematopoietic Stem Cell Transplantation". It was an honor to contribute to this work, which addresses a highly relevant topic and demonstrates significant potential for publication.

I have identified minor points for adjustment, such as providing more detailed discussions in specific sections, incorporating tables or figures to summarize key information, and refining the formatting for consistency. With these suggested revisions, I believe the article will fully meet the journal's standards and make a meaningful impact on the scientific community.

Minor Revisions

The text requires a review of acronyms, which should be written in full when first mentioned in the text. For example:

Line 38: natural killer (NK)

Line 51: graft-versus-host disease (GVHD)

Line 83: tumor necrosis factor-alpha (TNF-α), interleukin-6 (IL-6)

Line 92: regulated on activation, normal T cell expressed and secreted (RANTES)

Line 159: killer-cell immunoglobulin-like receptor (KIR)

This adjustment should be applied consistently throughout the manuscript.

  1. Thank you for your comments, we have made the corrections

Major Revisions

I believe the authors could enhance the manuscript by:

Adding Tables/Graphs: Summarize key data (e.g., success rates of different cell therapies) in tables or graphs to improve visualization, particularly for topics like the use of NK cells.

  1. The tables were added

Including a Future Perspectives Section: Add a specific section outlining challenges and future directions to enhance the clinical applicability of the findings. This would provide valuable insights for advancing the field.

  1. We have added information about Future Perspectives

With these revisions, the manuscript would further solidify its contribution to the literature and improve its clarity and impact.